# Direct observations of airflow separation over ocean surface waves

**Marc P. Buckley** [1] ✉, **Jochen Horstmann**[1,3], **Ivan Savelyev**[2,3] & **Jeff R. Carpenter** [1,3]

A large portion of the kinetic energy found within the ocean originates from the growth of ocean surface waves under the action of wind. However our understanding of wind wave dynamical coupling mechanisms remains incomplete. Competing theories exist but direct observational evidence is lacking, due to the technical challenges involved in measuring wind and wave dynamics in the vicinity of the highly energetic wavy ocean surface. Here, direct observations of airflow dynamics in the first millimeters to meters above ocean surface waves are shown. These were achieved using laser imaging techniques on the Floating Instrument Platform FLIP in the Pacific Ocean. The results show that two dynamical wind-wave coupling regimes coexist. Short (~1 m wavelength), strongly wind-forced waves travel more slowly than the wind and cause intermittent airflow separation events. On average, these slow waves are coupled with the airflow via a sheltering mechanism, while longer (~100 m), faster waves induce orbital motions in the airflow.

The exchanges of momentum, energy, and mass between the atmosphere and the ocean, are strongly influenced by small scale physical processes very close to the ocean surface, within the first centimeters of the coupled oceanic and atmospheric wave boundary layers[1]. A significant portion of the kinetic energy within the ocean originates from atmospheric forcing, in particular through the generation and growth of ocean surface waves under the action of wind[1,2]. Wind wave growth results from a difference in airflow pressure between the leeward and windward sides of the wave[3,4]. Yet the mechanisms that trigger this pressure perturbation, remain the subject of debate[1,5,6]. Physical interpretations are largely based on one of two competing theories, "critical-layer theory"[7], a linear shear-flow instability mechanism that takes place within the critical layer (thin region in the airflow, where the mean airflow speed matches that of the waves), and the "sheltering" mechanism, a turbulent process analogous to turbulent boundary modulations over low hills[8], whereby slow, wind-forced waves create a sheltered zone (from the wind) on their lee-side[9].

Field observations[4,10,11] and high resolution laboratory measurements[12] suggest that waves traveling faster than the wind very near the surface, but slower than the wind at ~10-m height, receive a significant amount of energy through the critical layer mechanism.

However, recent laboratory measurements show frequent, highly turbulent airflow separation or "separated sheltering"[13] events over strongly forced wind waves[9]; these are incompatible with linear critical-layer theory. On the other hand, longer waves that travel faster than the wind, are expected to cause sheltering effects on their windward side[8] (though observational evidence is scarce), and very fast-running waves (swells) can return momentum back to the atmosphere in the form of wave-induced winds[14–18].

When the wind forcing increases, wave steepening, breaking, and related turbulent processes are associated with an increase in air-sea exchanges of momentum and scalars[1,19–23]. In high to very high wind conditions, momentum and $CO_2$ fluxes across the ocean surface eventually stop increasing[24–28]. Airflow separation past wave crests has been suggested as a possible reason for this flux saturation[25,28,29]. Yet direct, in situ observational evidence of airflow separation is lacking, due to difficulties in measuring airflow within centimeters of the wavy ocean surface.

We tackle these open questions on the structure of the airflow over the broad spectrum of ocean waves through the utilization of laser imaging techniques that capture airflow kinematics over ocean waves. Here we show in situ, instantaneous two-dimensional snapshots

[1]Institute of Coastal Ocean Dynamics, Helmholtz-Zentrum Hereon, Geesthacht, Germany. [2]Remote Sensing Division, U.S. Naval Research Laboratory, Washington, DC, USA. [3]These authors contributed equally: Jochen Horstmann, Ivan Savelyev, Jeff R. Carpenter. ✉e-mail: marc.buckley@hereon.de

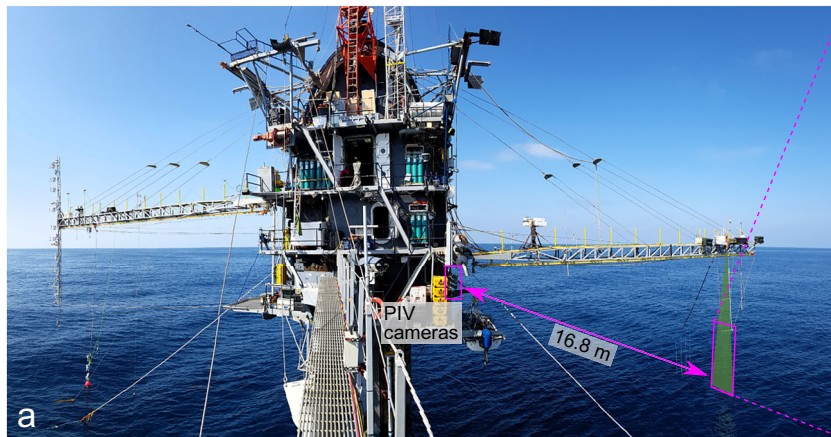
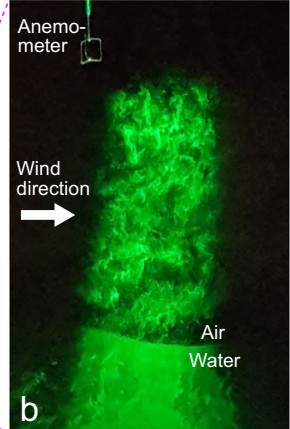

**Fig. 1 | Airflow Particle Image Velocimetry (PIV) over the Pacific Ocean from the Floating Instrument Platform FLIP.** A vertical array of PIV cameras captured the motions of the airflow just above the waves that traveled underneath the tip of FLIP's starboard boom (**a**). A high-power green laser sheet illuminated airflow seeding particles (fog) within the turbulent airflow over the air-sea interface (**b**).

of airflow velocities in the first millimeters to meters above ocean surface waves, acquired over the Pacific Ocean, from the Floating Instrument Platform FLIP, a unique research platform designed for the study of air–sea interaction (Fig. 1). The data were collected in October 2017 over the ocean ~50 km offshore of Santa Monica, California, where the water depth is about 1000 m. A large field of view (0.69 m × 2 m), high resolution (up to 1 airflow velocity vector every 1.2 mm$^2$), laser imaging system was developed and installed on the platform. It consisted of a vertical array of five cameras mounted on the first deck, that were focused on the airflow above waves that passed beneath the tip of the starboard boom (Fig. 1). A high-power, pulsed laser created a green light sheet down to the wavy surface, thereby illuminating the air-water interface, as well as fog micro-droplets that seeded the airflow. These flow seeding particles were injected into the airflow upwind of the imaging region. The system, operated at night, captured snapshots of the green laser-illuminated particles (Fig. 1, panel b), the motions of which were used to quantify the airflow kinematics within the first millimeters to meters above the wavy ocean surface, in a technique known as Particle Image Velocimetry (PIV).

We are able to directly observe and quantify turbulent, intermittent airflow separation events over short wind waves (with wavelengths ~1 m). These slow waves—with respect to the local, near surface wind speed—ride upon longer waves (~100 m) that travel faster than the wind. Using conditional averaging with respect to the along-wave position (wave phase), we demonstrate the coexistence of two different dynamical wind-wave coupling mechanisms: the slow wind waves are coupled with the airflow via a sheltering mechanism, while the fast waves induce orbital motions in the airflow.

## Results and discussion

### Airflow separation events

Instantaneous, two-dimensional snapshots of the velocity field within the first 50 cm of airflow above ocean surface waves, reveal the turbulent structure of the airflow as it is dynamically coupled with the waves (Fig. 2).

The bulk of the airflow shown in Fig. 2a (horizontal velocity component) is traveling significantly faster (at ~4 m/s) than the wave, which is moving with a crest speed of ~1.4 m/s. This velocity difference, combined with the geometry of the wave (with a half wavelength $\lambda/2$ ~ 0.56 m and a crest-to-trough height of $H$ ~ 9.4 cm), causes the local airflow boundary layer to detach from the surface just past the crest of the wave, creating an airflow separation (or separated sheltering) event, whereby a region of the airflow just downwind of the wave crest

is spared from the full force of the wind. In fact, the sheltered region is characterized by a slight flow reversal (Fig. 2a, b). In panel b, vertical profiles of the horizontal velocity field at the crest and downwind of the crest illustrate the strong modulation of the airflow's boundary layer by the wave.

Two-dimensional instantaneous vorticity fields (spanwise component, Fig. 2e–h) display typical boundary layer separation features[30]. The near-surface high vorticity layer (due to viscous shear, ref. [31]), relatively thin upwind of the crest, thickens past the wave crest, and is ejected away from the surface. The flow reversal (or "recirculation", see for example ref. [30]) is accompanied by a patch of negative vorticity (panel f). Downwind of the detached region, we observe a gradual regeneration of the surface vorticity layer. Such vorticity patterns, previously only observed in controlled laboratory studies, are characteristic of airflow separation events over slow (with respect to the wind speed above), strongly forced wind waves[9,32].

We note that the airflow separation event takes place on a wave that is not very steep ($H/\lambda$ ~ 0.1). This, combined with its wave age ($c/u_*$, where $c$ is the wave crest speed and $u_*$ the friction velocity) of 6.4, means that this wave is far from the "incipient breaking" limit suggested by Banner and Phillips[33], and falls within the range of waves that have a 30% chance of experiencing airflow separation, following the classification of Buckley et al.[9]. Nonetheless, the airflow separation event persists for the duration of the imaging interval of the slow wind-wave displayed in Fig. 2 (0.27 s, with one velocity field every 1/15 of a second), but the rapid evolution of the velocity and vorticity fields demonstrates the highly variable and turbulent (or "intermittent", ref. [30]) nature of the event.

These instantaneous observations of airflow velocities above such short, slow waves, indicate that the momentum and mechanical energy transfers from the atmosphere to these waves are dominated by sheltering dynamics. In fact, the observed sheltering event (Fig. 2a) is incompatible with a linear (critical layer) growth mechanism because the contour $u = c$, where $u$ is the horizontal velocity and $c$ the wave's crest speed (black dashed lines in Fig. 2a), merely reflects the separated flow, rather than a small perturbation to the mean wind profile.

### Instantaneous structure of the airflow over fast waves

The ocean surface is generally composed of a broad spectrum of waves of different scales and ages (e.g.,[4,16,19]). During this experiment, relatively long ($\lambda$ ~ 70 m) and fast waves were observed, traveling at a peak phase speed of 10.4 m/s, with a wave age of $c/u_*$ = 33 (or $c/U_{10}$ = 1.2, where $U_{10}$ is the mean wind speed at a height of 10 m above the water surface), upon which smaller, slower locally generated wind waves

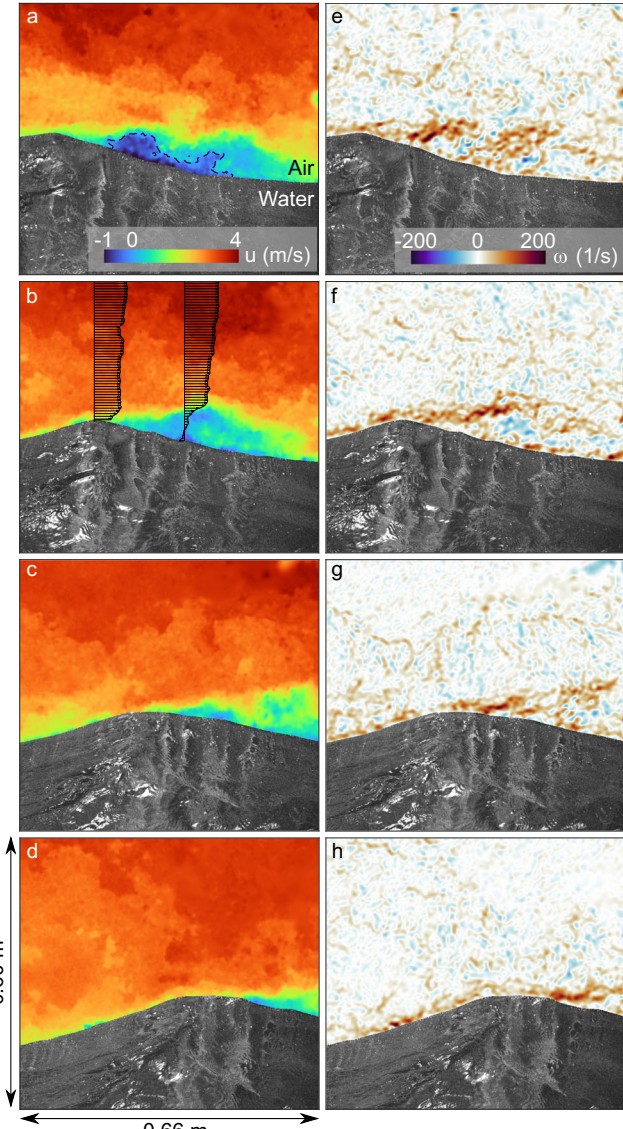

**Fig. 2 | Airflow separation events over slow ocean surface waves, measured by Particle Image Velocimetry (PIV).** The velocity fields (horizontal component $u$, **a–d**) are in a frame of reference traveling at wave crest speed ($c = 1.4$ m/s). The dashed line indicates where $u = c$ for the first PIV snapshot (**a**). The vorticity fields (spanwise vorticity $\omega$, clockwise positive **e–h**) demonstrate the detachment of the surface viscous shear layer past the crest of the wave. The velocity resolution is 1 velocity vector every 1.2 mm². The snapshots are taken every 1/15 of a second (66 ms). The age of this wave (with a wavelength ~1 m) is $c/u_* = 6.4$ or $c/U_{10} = 0.22$, which classifies it as a "slow" wave with respect to the wind speed blowing over it[8]. It is riding on the crest of a longer wave (with a wavelength ~ 70 m). $U_{10}$ is the mean wind speed at a height of 10 m above the water surface, and $u_*$ is the friction velocity, a measure of the total air–sea momentum flux.

such as the one displayed in Fig. 2 were superimposed. A reconstructed snapshot of the airflow velocities over a long, fast wave (Fig. 3) contrasts with the airflow kinematics over the short, slow wave from Fig. 2. Contrary to the slow wave case where the sheltered region is to the right of the crest (leeward side with respect to the mean wind), the magnitude of the horizontal velocity (panel a) decreases here on the left side of the wave, which is traveling faster than the local, near-surface wind. In other words, a form of sheltering takes place on the windward face of the long, fast wave's crest (also known as "negative asymmetry"[13]). The vertical velocity field (panel b) indicates that the orbital motions of the wave draw (resp. push) the airflow downward

(resp. upward) on the windward (resp. leeward) side of the crest. The vectors plotted in panel c represent the mean velocities across the sampled air column; they illustrate the forcing of the airflow by the fast wave's orbital motions. This quasi-instantaneous observation of the modulation of the airflow by a single, fast wave over the open ocean, is in qualitative agreement with previous laboratory studies of instantaneous airflow kinematics over waves of similar age (see for example ref. 31, who conducted an experiment with a similar wave age, $c/u_* = 31.7$).

## Wave-induced motions

In order to quantify the average effect of the waves on the airflow, and better understand the airflow's coupling with the slow waves versus the fast waves, the airflow velocity horizontal and vertical components ($u, w$, respectively) were binned and averaged according to their along-wave location, or phase $\phi$ (phase averaging). Subsequently subtracting the mean velocity profile from the phase average provides wave-coherent motions, or velocity perturbations induced by the waves:

$$\tilde{u}(\phi, z) = \langle u \rangle(\phi, z) - \overline{u}(z) \qquad (1)$$

where $z$ is a measure of the vertical distance from the water surface, $\langle u \rangle$ and $\overline{u}$ represent the conditional (with respect to wave phase) and temporal averages, respectively (e.g., Sullivan et al.[34], Hara and Sullivan[35]).

The mean wave-coherent velocities ($\tilde{u}, \tilde{w}$) shown in Fig. 4 demonstrate the important differences between wind-wave coupling mechanisms for slow, locally generated wind-waves (with wavelengths $\lambda < 1$ m, Fig. 4a, b) versus those for fast waves (with a peak wavelength of ~70 m, Fig. 4c–f), as already suggested by the instantaneous observations (Figs. 2 and 3). The slow waves are, on average, causing the relatively high velocity wind above, to accelerate and move upward as it approaches crests, and to decelerate (sheltering) and move downward past the crests (panels a and b). We observe a phase shift of ~0 (resp. $-\pi/2$) between the maximum horizontal (resp. vertical) airflow velocity and the water surface elevation in these slow waves. We note that intense separated sheltering events (e.g., Fig. 2) are not directly represented in the phase average. This is likely due to the intermittency of airflow separation events[30], that occur over a portion of the spectrum of waves[9]. Therefore the observed mean wave-coherent pattern points toward a dominance of the "nonseparated sheltering"[13] mechanism for momentum and energy flux into the slow waves. The wave-coherent perturbations of the horizontal velocities are strongest within the "inner region", defined by Belcher and Hunt[13] as a thin region close to the surface where the waves are expected to induce perturbations of the turbulent stress. These produce a sheltering effect that modulates the surface pressure, which, in turn, may lead to the growth of wind waves[13]. These in situ observations of wave-induced airflow velocities within the first millimeters above slow wind waves (with estimated wave ages $c_p/u_*$ between 2.5 and 4.2) are in good agreement with past laboratory[31,36] and numerical[34] studies. There, similar phase shifts between wave-induced perturbations and water surface elevation were observed and modeled, albeit in simplified wave conditions (short fetch and monochromatic, respectively). Notably, the presently observed phase shifts match closest with the numerical reference case $c_p/u_* = 0$ of Sullivan et al.[34], for flow over a solid wavy surface. This further supports the hypothesis that the relatively fast airflow is decoupled from the slow waves' orbital motions, leading to sheltering as the dominant coupling mechanism.

Using the non-separated sheltering growth rate parametrization suggested by Belcher and Hunt[8], we can express the non-dimensional

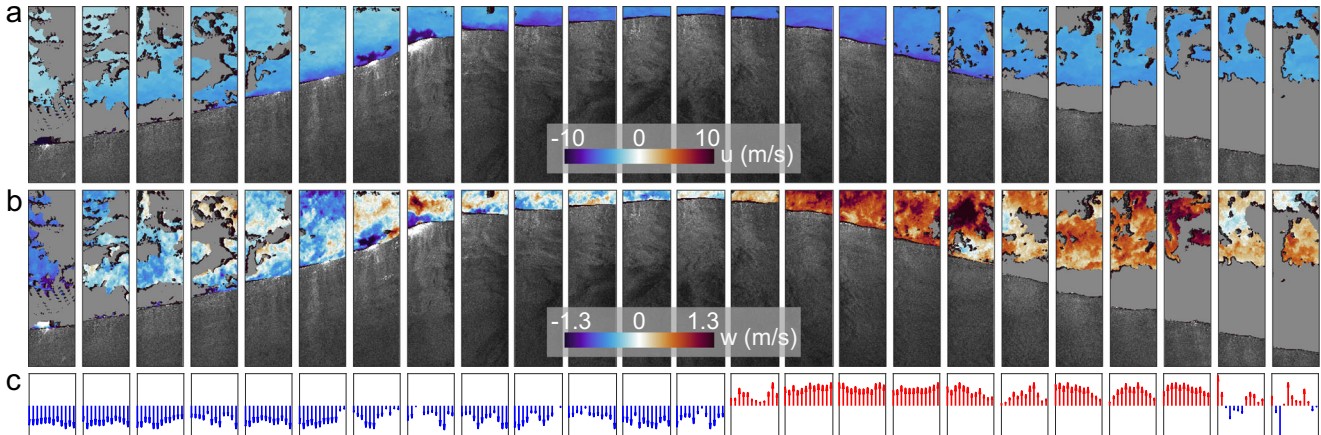

**Fig. 3 | Reconstructed instantaneous airflow velocity field measured by Particle Image Velocimetry (PIV) over a fast ocean surface wave.** Both the mean wind, and the wave, with a wavelength ~ 70 m, are propagating from left to right. **a** Horizontal velocity component $u$, in a frame of reference traveling at peak crest speed (10.4 m/s), **b** Vertical velocity component $w$. Each of the 24 PIV snapshots is 0.69 m wide, and 2 m high, with 1 velocity measurement every 0.12 mm². The individual snapshots were taken every 1/15 s; time increases from right to left. **c** Bin-averaged vertical airflow velocity, with blue indicating downward and red upward. Each bin is 6 cm wide and spans the entire height of the PIV image. The mean wind speed at 10 m height was $U_{10} = 8.3$ m/s.

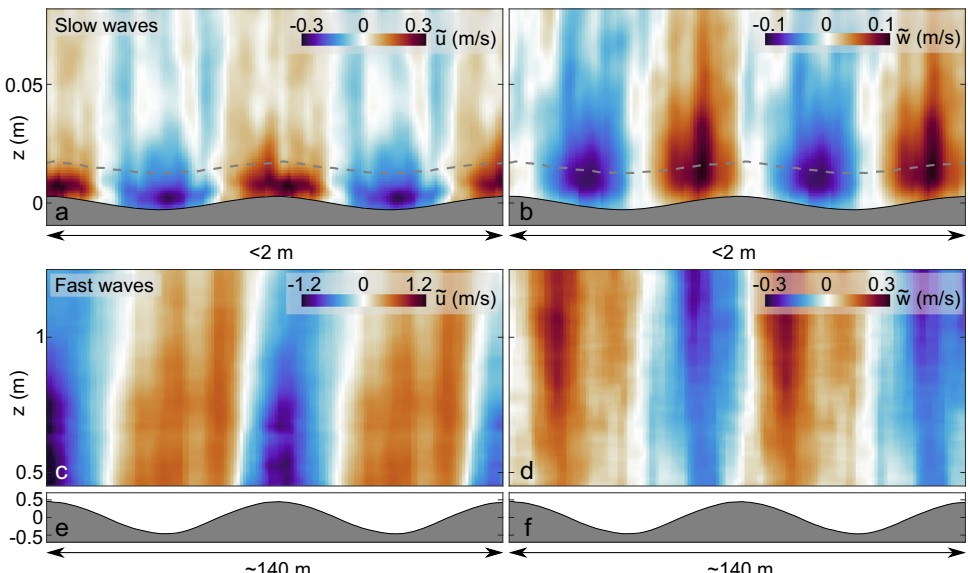

**Fig. 4 | Mean wave-coherent airflow velocities.** Horizontal ($\bar{u}$) and vertical ($\bar{w}$) velocity components are shown. Slow waves (**a**, **b**) and fast waves (**c**, **d**) induce broadly differing patterns in the airflow. The phase-averaged water surface elevation is shown below the velocity fields (gray shading) for the slow waves, and in (**e**) and (**f**) for the fast waves. The gray dashed line represents the height $L_i$ of the "inner region", which satisfies $kL_i(U(L_i) - c) = 2\kappa u_*$, where $\kappa$ is the von Karman constant, $c$ the peak phase speed of the waves, $k$ the wavenumber, and $U(z)$ the mean wind velocity profile (Belcher and Hunt[8] (their equation 26, see also Grare et al.[10], Buckley and Veron[31]).

wave energy growth rate $\beta$ (e.g., Komen et al.[37]), as:

$$\beta = 4\pi r \frac{U^4(\lambda)}{U^4(L_i)} \left(\frac{u_*}{c}\right)^2, \qquad (2)$$

where $L_i$ is the height of the inner layer, $U$ the mean wind profile in a frame of reference moving with the waves, and $r$ the air–water density ratio. We find a non-dimensional nonseparated sheltering growth rate $\beta$ between 0.015 and 0.102 (computed by using values of $\lambda$ between 0.35 and 1 m). This value agrees well with historical observations of wave growth rates, as shown in Fig. 5, where the dimensionless wave growth rate $\beta$ is represented as a function of inverse wave age $u_*/c$. This agreement is consistent with the growth of the observed slow waves by the sheltering mechanism.

The fast waves (Fig. 4c, d) force orbital motions in the airflow, with phase shifts approaching $\pi$ (horizontal velocity component) and $\pi/2$ (vertical component) with respect to the phase averaged water surface elevation (shown in panel e). The water orbital velocities of the fast waves are expected to be ~0.5 m/s, as estimated from linear theory. This value is consistent with a strong coupling of the water orbital motions with the measured wave-coherent airflow motions, which reach values of ~1 m/s for $\tilde{u}$ and ~0.3 m/s for $\tilde{w}$ (panels c and d). This result is in general agreement with past single-point observations[4,10,11], and numerical simulations[16,34]. The fast waves observed here, as they are traveling ~1.9 m/s faster than the wind at 10 m height, are likely not influenced by dynamics near their critical layer, which, in this case, would be at a height ~100 m[8].

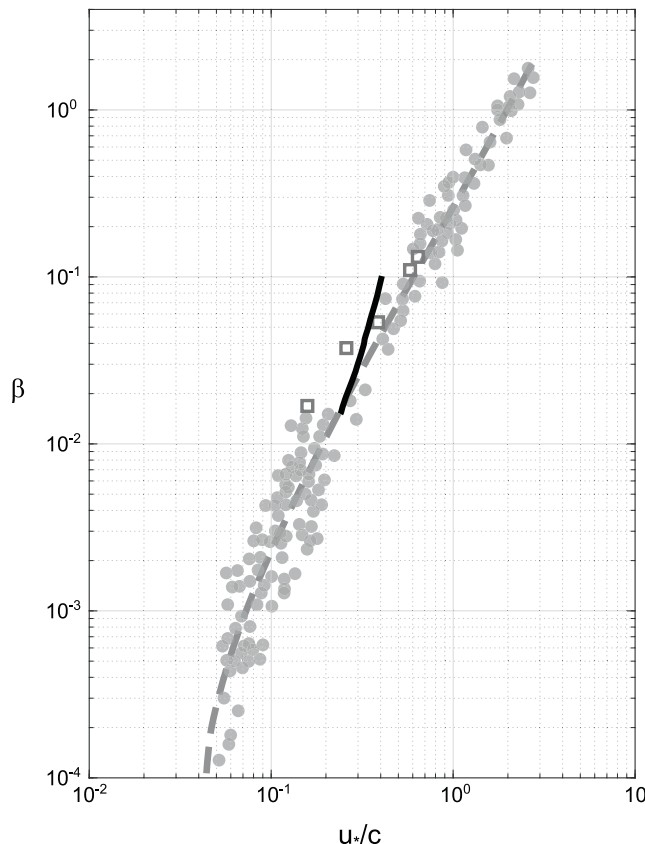

**Fig. 5 | Wave growth rate through the sheltering mechanism.** Dimensionless wave growth rate $\beta$, as a function of inverse wave age $u_*/c$. Gray squares represent wave growth estimates from recent laboratory measurements[9]. Gray circles show past laboratory[48] and field[49] observations compiled by Plant[50], while the dashed gray line represents the growth rate given by Miles' linear theory[7] (see Komen et al.[37], Janssen[51]). The black line is the slow wave "nonseparated sheltering" growth rate[13] from the current study (Eq. (2)), for wavelengths between 0.35 and 1 m.

## Insights and future steps

The present study shows direct, detailed observations of the airflow's modulations over short ($\lambda \lesssim 1$ m), slow waves ($c/u_* \leq 4.2$), and longer ($\lambda \sim 70$ m), fast waves ($c/u_* = 33$), within the first millimeters to meters above the ocean surface. The measurements offer access to turbulent processes in the airflow above ocean surface waves (e.g., separated and non-separated sheltering events), as well as to mean, wave-induced airflow perturbations. We find that there are different wind-wave coupling mechanisms acting simultaneously on a realistically broad, in situ ocean wave spectrum. This is demonstrated by direct observations of two different members of this spectrum. Fast waves are coupled with the airflow above, through their linear orbital motion, whereas slow waves show a coupling through separated (airflow separation) and non-separated sheltering, within the first centimeters above the water surface. Understanding these rapidly varying, turbulent, wave-induced sheltering events is key to quantifying the exchanges of momentum, energy, heat and mass between the atmosphere and the ocean. This is especially true in high to very high wind conditions, where slow (young), strongly forced airflow separating waves are ubiquitous. Revealing these mechanisms of wind-wave growth, can require a millimeter to centimeter scale resolution of the airflow adjacent to the ocean surface. For intermediate wave ages, (e.g., for $14 \lesssim c/u_* \lesssim 25$[8]), the competing dynamical roles of turbulent (e.g., sheltering) versus linear (e.g., Miles' critical layer theory,[7]) mechanisms, remain a topic of active research. New methods are being developed, to quantify these processes from our high resolution airflow observations, within

this previously inaccessible region of the atmosphere-ocean boundary layer.

## Methods

### In situ large field of view Particle Image Velocimetry (PIV)

**Instrumentation.** A large field of view, high resolution PIV system was developed specifically for this study of airflow dynamics above ocean surface waves from FLIP. The system was composed of the following elements. Seeding particles (fog droplets, with a diameter of ~15 µm) were injected into the airflow by pumping local seawater into an array of spray nozzles, located ~3 m upwind of the imaging region. A vertical array of five, 5 Megapixel USB3 CMOS cameras (Grasshopper 3, Point Grey Research) was installed on the first deck of FLIP, each of which was fitted with a remote lens controller for focus and aperture control (developed in-house, using an Arduino microcontroller). The top camera (large field of view), fitted with a Canon EF 50 mm lens, imaged the entire PIV region, while each of the remaining 4 cameras, fitted with a Canon EF 200 mm lens, imaged only a portion of the field of view, in order to achieve the desired resolution (300 µm/pixel). The images were stitched together prior to computing velocity vectors from the particle displacements. A downward-facing pulsed green (532 nm) Nd:YAG laser (Quantel Evergreen, 200 mJ/pulse) was positioned above the PIV field of view. PIV cameras and laser were controlled by National Instruments counter/timer hardware (PCIe 6612) connected to a computer located in the lab on the 3rd deck of FLIP. PIV measurements took place at night, in order to ensure that the only light source was the PIV laser, and thus minimize optical noise on the PIV images. The PIV acquisition was performed for intervals of 70 s, every 10 min, to allow for transfer of the buffered images from the computer memory to the hard drives. Each 70 s dataset represents a total of 1050 2D velocity fields, each with up to 6776 × 2301 measured velocity vectors, or a total of 2,416,050 (vertical) velocity profiles.

**Imaging.** As the laser light penetrates the water surface, the intersection of the laser sheet with the water surface delivers a high resolution (300 µm/pixel) picture of the instantaneous geometry of the water surface (upper boundary of the gray scale region in Fig. 2). The intensity variations below the interface are caused by the refraction of green PIV laser light through the rough water surface, and are as such a function of the local variations in the slope of the water surface [see also refs. 38,39].

The suitability of the seeding particles to follow the turbulent motions of the airflow was assessed by computing the Stokes number $St$ of the fog droplets. We find $St = \tau_p/\tau_k \ll 1$, where $\tau_p = 1.2$ ms is the Stokes time scale of the largest particles, and $\tau_k = 23.6$ ms is an estimate of the Kolmogorov time scale, estimated using the measured airflow friction velocity and a length scale of 1 m (cutoff wavelength for the slow waves)[40].

One of the challenges with in situ (outdoor) airflow PIV is the difficulty to achieve high seeding particle density within the entire field of view of the cameras. Some of the raw PIV images inevitably contained patches with low seeding density. An image variance based algorithm was developed and used to remove these patches automatically before computing PIV velocity fields. Low seeding density was observed predominantly over the troughs of peak (fast) waves with wave heights ~1 m. Hence the very near surface airflow measurements presented in this paper are mostly acquired above the crests of the fast waves. Therefore the results presented for slow wind waves are valid for waves riding on the crests of the fast waves.

**Procedure.** Prior to starting the PIV measurements, the dominant wind direction was monitored live using a sonic anemometer. Once the wind direction was approximately aligned with the laser sheet (direction perpendicular to the starboard boom), the PIV data acquisition started. This procedure ensured the presence of seeding particles within the

laser light sheet, and that the PIV technique captured the dominant wind velocity components.

## Environmental conditions and supporting measurements

The PIV images for Fig. 2 were acquired on 17 October, 2017 at 10:27 pm, local time. The local 10-m wind speed was 6.6 m/s. The data for Figs. 3 and 4 were acquired on 20 October, 2017, at 7:13 pm. The local wind speed was then 8.4 m/s. Short wind waves (with wavelengths ~1 m) were typically traveling in the direction of the local wind. Alignment of the peak wave travel direction with the local wind direction was checked using directional wave field information provided by a waverider buoy moored within 1 km of FLIP's location.

A suite of sonic anemometers (including the one visible in Fig.1b, just above the PIV imaging region) and wave gauges provided additional, single-point, fixed height wind-wave measurements, co-located with the PIV imaging region. Two 3D anemometers (Gill Windmaster) measured wind velocities with a sampling frequency of 20 Hz at fixed heights above the mean water level (2.6 and 4.6 m, respectively). A weather station (measuring 2D wind speed, temperature, relative humidity, atmospheric pressure) was positioned at a height of 10.6 m above mean water level. The latter measurement was used to estimate the friction velocity ($u_*$), needed for the calculations of wave age and growth rate, via the COARE bulk flux algorithm (Edson et al.[19], Fairall et al.[41,42]). During the same time period, Ortiz-Suslow et al.[43] also performed wind speed measurements (vertical profiles of the wind speed), on the port boom of FLIP. Their observations point to the existence of a constant flux layer at the time of our measurements, and are in good agreement with our total momentum flux estimates[43]. Two down-looking distance sensors were used as wave gauges (sonic and LIDAR, respectively), acquiring time series of the water surface elevation at a spanwise distance of 60 cm from the PIV field of view, and with a sampling frequency of 20 Hz. A motion sensor (VectorNav VN-100 Rugged IMU) was placed next to the PIV laser on FLIP's starboard boom, and acquired the motions of FLIP at a frequency of 20 Hz. A frequency spectrum of the water surface elevations obtained from different methods, at the location of the PIV measurements is provided in Fig. S1 in the supplementary material. The water surface profiles obtained from the PIV images provide two data products. First, any column of the images can be used as a single point wave gauge, yielding a time series (with a sampling frequency of 15 Hz, the PIV pair acquisition frequency) of the water surface elevation. Here, the middle of the PIV field of view was used (black curve). Second, wavenumber spectra can be computed from the spatial surface snapshots of each PIV image (0.69 m wide, with a resolution of 300 µm/pixel); these were subsequently averaged yielding the purple curve.

A cross-spectral analysis (Fig. S2) of the sonic anemometer wind speed measurements ($u_a$, $w_a$) with the water surface elevation measurements ($\eta$), co-located with the PIV measurements, supports the phase shift observations from the phase-averaged PIV fields for the fast waves in Fig. 4c, d. By computing the cross-spectral densities between the horizontal (resp. vertical) airflow velocity components and the water surface elevations, denoted $S_{u_a\eta}$ (resp. $S_{w_a\eta}$), we find that the coherence function $\gamma_{u_a\eta} = |S_{u_a\eta}|^2/S_{u_au_a}S_{\eta\eta}$ (resp. $\gamma_{w_a\eta} = |S_{w_a\eta}|^2/S_{w_aw_a}S_{\eta\eta}$) is maximum at the peak wave age ($c/u_* = 33$). At this wave age, their respective phase shifts ($\phi_{u_a\eta} = \arg(S_{u_a\eta})$, $\phi_{w_a\eta} = \arg(S_{w_a\eta})$) are -0.9$\pi$ and 1.2$\pi$/2. This result supports the hypothesis that (linear) wave orbitals induce the wave-coherent motions in the airflow, in accordance with linear theory, which predicts phase shifts of $\pi$ and $\pi/2$, respectively[4].

## Wave phase detection and averaging

For the fast waves, instantaneous wave phases were defined as the argument of the Hilbert transform of the water surface elevation time series, for each along-wind ($x$-direction) location in PIV snapshot $j$, $\phi_j(x) = \arg[\mathscr{H}\{\eta_j(x)\}]$, where $\eta(x)$ represents the surface elevation at

along-wind position $x$ (see also Buckley and Veron[31], Melville[44]). An example of wave phase detection from Hilbert analysis is shown in Fig. S3b, applied to a time series of surfaces detected on the PIV images, and to sonic wave gauge data (panel a). Here, each point of the time series of "PIV surfaces" is the middle of the detected surface within each PIV image. Panel a shows the relatively good agreement between the wave gauge and imagery water surface elevation measurements. This agreement also holds for the phase averaged water surface elevations (Fig. S4c). In addition, we checked the validity of the PIV velocity measurements by comparing the airflow velocities obtained at the top of the PIV field of view (at a height of $z = 1.5$ m above the mean water level) with the airflow velocity observations from the lowest sonic anemometer ($z = 2.6$ m). Figure S4a, b shows the phase averages of these quantities (horizontal and vertical components, respectively).

In the case of the slow waves, the spatial water surface elevation profile from each PIV snapshot was first high-pass filtered to isolate the slow waves (with a cut-off wavelength of $\lambda = 1$ m). A Hilbert transform was then applied to the filtered profiles, yielding a wave phase estimate for each point along the imaged water surface[40]. The velocity fields were mapped from the cartesian frame of reference ($x$, $z$) to a surface-following frame of reference ($x$, $\zeta$), where $\zeta = z - \eta(x, t)$ (as was previously done in laboratory studies, for example by Grare et al.[45], Tenhaus et al.[46]). For the comparatively more regular, fast waves, the phase averaging was performed in the cartesian frame of reference.

**Platform motion.** The airflow velocity and surface displacement measurements presented here were corrected to account for the motions of the platform, and projected into earth coordinates. This was achieved by using a combination of motion sensor measurements[47], water surface imagery on the PIV images, as well as images of the lowest anemometer captured by the large field of view PIV camera.

## Data availability

The data corresponding to the results presented in this study have been deposited in the Figshare database https://doi.org/10.6084/m9.figshare.29135960.

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

## Acknowledgements

M.P.B. acknowledges support from the US National Science Foundation (grant AGS-PRF-1524733). M.P.B and J.R.C. acknowledge support from the Deutsche Forschungsgemeinschaft (DFG, German Research Foundation, project number 274762653, Collaborative Research Centre TRR 181 "Energy Transfers in Atmosphere and Ocean"). I.S. acknowledges support from ONR-NRL base program WU 73-1Y91. We thank Burkard Baschek, Ruben Carrasco-Alvarez, Lucas Merckelbach, Yoana Voynova, Qing Wang, Dave Ortiz-Suslow, Ryan Yamaguchi, FLIP's captain Tom Golfinos and crew, as well as the Office of Naval Research (USA), and the Helmholtz Association (Germany).

## Author contributions

M.P.B. and J.H. conceived the study and designed the experimental setup. M.P.B and I.S. further developed and installed the measurement setup, and acquired the data on the Research Platform FLIP. M.P.B., J.H and J.R.C. shaped the data analysis and interpretation. M.P.B. conducted the data analysis, directed and composed the manuscript. All authors discussed the results and edited the manuscript.

## Funding

## Competing interests

The authors declare no competing interests.
