## [Transparent Peer Review file · Nature Communications]

Direct observations of airflow separation over ocean surface waves

Corresponding Author: Dr Marc Buckley

Version 0:

Reviewer comments:

Reviewer #1

(Remarks to the Author)

In this study, the authors used a PIV system installed on the FLIP ocean research platform to measure ocean wind turbulence. The authors observed that the boundary layer separated for relatively short-wavelength, “slow” component waves, and argued for the importance of the sheltering mechanism in wind wave development. The successful PIV measurement of oceanic wind turbulence in actual sea areas suggests the possibility of new research developments in the future, and the academic value of this work is high. In particular, if the separation of the boundary layer plays an important role in wind wave development, it is even possible that it will overturn the previous theoretical research that explains wind wave development. The reviewer highly evaluates the novelty and academic value of this paper, and therefore evaluates this paper as Minor Revision.

The method of measurement using PIV is unclear.

In contrast to tank experiments, the difficulty of PIV measurement in real sea areas is to set the downwind direction appropriately for wind waves with directional propagation. Therefore, in order to achieve continuous PIV measurement, it is necessary to monitor the wind direction in real time and adjust the direction of laser beam irradiation as needed. In this paper, there is no description of how the downwind direction was set and the measurement was performed. This is an unclear point regarding the in-situ PIV method, so I recommend that it be described. Since the measurement time is only 70 seconds, will the data analysis results only be shown if the irradiation direction of the laser sheet coincides with the downwind direction?

In order to calculate the growth rate of the wind wave, it is necessary to define the friction velocity. Firstly, is a constant flux layer confirmed? If a data sample is in the 70s, is statistical representativeness guaranteed? And how has it been confirmed?

Comparison with previous research

In order to make the validity of the PIV measurements more robust, it is recommended to compare the results of this study with the results of previous research. For example, Sullivan et al. (2000, JFM) show the turbulent intensity and Reynolds stress distribution at the wavy boundary obtained by DNS. It would be better to show to what extent the turbulent characteristics obtained in this study agree with them.

As a conclusion of this study, it is difficult to understand the position of the Miles mechanism (1958) and the sheltering mechanism proposed by Jeffreys (1925) in the development of wind waves. While it is stated that the contribution of the sheltering mechanism to the wind wave growth rate is large, the importance of the Miles mechanism is claimed in the critical layer of a few millimetres.

Reviewer #2

(Remarks to the Author)

Review of Buckley et al, Direct observations of airflow separation over ocean surface waves

This paper presents field measurements of the airflow separation over short and long waves during a campaign with FLIP outside the coast of California. The results point to two regimes for how surface waves are coupled to the airflow. The short waves are observed to experience a sheltering mechanism, whereas the long (70m and more) affect the airflow. The results

are important and the reporting is timely since the general problem of wave growth is one which has direct implications for numerical weather prediction and climate modeling. The problem remains far from solved, but there is general agreement that a critical layer theory is the most efficient mechanism by which momentum is transferred between the atmosphere and the wave field. The authors are thus presenting a somewhat contrarian view, in that the sheltering seems to be quite important. They also claim to observe a direct momentum transfer from the long waves to the lower atmosphere. This latter mechanism has been studied experimentally (see Semedo et al, 2009), and is potentially of importance for swell attenuation. The middle ground, however, is not well covered by this study. That is, we are presented with very long waves that outrun the local wind and very short waves that are strongly forced by the wind. The critical layer theory is supposedly filling in the intermediate scales, but the authors state that the critical layer falls either too close to the wavy surface to be resolved for their short waves or is too high up in the case of long waves. This is a weakness of the study since the community would benefit from knowing how well the critical layer theory holds up in such experiments. We acknowledge that the extent to which this is feasible may be limited by the experimental setup, but we would encourage the authors to look for evidence of this middle ground where critical layer theory should apply. Moreover, it is not entirely clear from the manuscript how long the experiments went on; this should be clarified in the revisions.

Having said that, the paper is well written, relevant and important, and we would recommend publication following a round of revisions.

Major comments and questions

The literature review should be expanded to set the current experiments and results in context. This could be done either in the introduction or in the supplementary materials.

Can the analysis be extended to also cover mid-range wavelengths for which the critical layer would presumably coincide with the range of the PIV measurements?

Specific comments by section

Introduction

(As above) More in depth information about existing theories and contradictions between competing theories would be helpful to set the context for the current analysis and its importance, especially since the journal (Nature Communications) has a wide interdisciplinary scope. Therefore many readers are unlikely to have a strong background in the subject. If word limits prohibit the introduction from being expanded significantly, a more in-depth review of the literature covering the critical layer and sheltering mechanisms could be added to the supplementary materials for interested readers.

The dataset was collected already in 2017. Is this the first paper to analyze the dataset, or have there been other publications using the same data?

2.1 Airflow separation events

Fig. 2: Does the green laser penetrate the water surface? Do the gray regions beneath the air-sea interface in Fig. 2 represent the water column under the surface, or is it a re-projection of the surface reflection?

Fig. 2: Is the example short wave riding the crest or trough of a long wave? One would assume the former based on the text in the Methods section, but it would be helpful to state this more clearly also in the main text. Are the axes rotated relative to the instantaneous long-wave slope?

2.2 Instantaneous structure of the airflow over fast waves

Fig. 3: Are both the wave crest and (mean) wind propagating from left to right? Is windward/leeward defined with respect to the crest velocity? This could be specified in the caption.

2.3 Wave-induced motions

How many individual slow and fast waves were used to generate the ensemble-averaged airflow profiles? What is the full duration of the experimental dataset?

Do the slow wave ensemble averages contain breaking waves? If not, how do (or would) the mean airflow profiles differ for breaking waves?

Eq. 1: How were the inner layer heights L_i determined or defined?

Fig. 4: (How) Does the long-wave modulation of the short waves affect the mean airflow patterns? Does the mean airflow of the slow waves cover all long-wave phases? Would the authors expect there to be a difference in the short-wave airflow at different long-wave phases due to long-wave modulation of the short waves (e.g. crest vs trough of long waves)?

Fig. 4: Alternate layout suggestion: Keep 2-column layout, but with the left column consisting of 4 subpanels in each row, and the right panel consisting of one high panel (current f panel). Left column: 4-panel column with c+d on top 2 rows of left column. Include the wave form in the same way as in a+b (so that all subpanels include 0 on the y axis). On 3rd and 4th row of left column, show current a+b panels, with e.g. a box showing the horizontal and vertical scales of the short waves vs the

long waves.

3 Conclusion

The authors summarize their results, which show that short/slow waves are coupled to a sheltering/separation response in the nearfield airflow, whereas the long/fast waves induce orbital wakes in the airflow. This result is essentially (redundantly) stated twice back-to-back. What is lacking for readers with basic (or limited) background in the subject is a clear statement of how the presented results change or update our understanding of the process of wave growth by wind. As stated above, this should preferably be explained more thoroughly in the introduction, but more details could be added to the Supplementary materials.

4.1 In situ Large field of view Particle Image Velocimetry (PIV)

Will the code for the PIV algorithm used in this study be made publicly available?

What level smoothing or filtering was performed on the raw PIV velocities vs. the final results shown in the manuscript?

4.2 Environmental conditions and supporting measurements

Are directional wave spectra available for the measurement campaign? These would be helpful for showing the relative directionality of the windsea and swell.

Will any of the data and/or code used in the manuscript be made publicly available?

References

Semedo, A, O Saetra, A Rutgersson, K K Kahma, H Pettersson (2009). Wave-induced wind in the marine boundary layer, *J Atmos Sci*, 66(8), pp 2256-2271, doi:10.1175/2009JAS3018.1

Reviewer #3

(Remarks to the Author)

Version 1:

Reviewer comments:

Reviewer #1

(Remarks to the Author)

The author's response to my questions was clear, and I have no further questions. This version is sufficient for acceptance of the paper.

Reviewer #2

(Remarks to the Author)

The authors have answered our concerns (this is a joint review by Rev #2 and Rev #3, an early career scientist). We commend the authors on a very nice study which, despite the challenges and limitations of the experiment, will surely make an important contribution to the ongoing effort to understand the fiendishly difficult problem of unraveling the processes responsible for wave growth.

Reviewer #3

(Remarks to the Author)

Response to the Reviewers

Reply: The authors would like to thank the reviewers for their constructive comments and suggestions that have helped improve the quality of this manuscript. The manuscript has undergone a thorough revision according to the reviewers' comments. Please see our responses below. For the reviewers' convenience, the reviewers' comments are written in black, our responses are in blue, and we have highlighted changes in the revised manuscript in blue.

Reviewer 1

In this study, the authors used a PIV system installed on the FLIP ocean research platform to measure ocean wind turbulence. The authors observed that the boundary layer separated for relatively short-wavelength, "slow" component waves, and argued for the importance of the sheltering mechanism in wind wave development. The successful PIV measurement of oceanic wind turbulence in actual sea areas suggests the possibility of new research developments in the future, and the academic value of this work is high. In particular, if the separation of the boundary layer plays an important role in wind wave development, it is even possible that it will overturn the previous theoretical research that explains wind wave development. The reviewer highly evaluates the novelty and academic value of this paper, and therefore evaluates this paper as Minor Revision.

Comment 1 The method of measurement using PIV is unclear. In contrast to tank experiments, the difficulty of PIV measurement in real sea areas is to set the downwind direction appropriately for wind waves with directional propagation. Therefore, in order to achieve continuous PIV measurement, it is necessary to monitor the wind direction in real time and adjust the direction of laser beam irradiation as needed. In this paper, there is no description of how the downwind direction was set and the measurement was performed. This is an unclear point regarding the in-situ PIV method, so I recommend that it be described. Since the measurement time is only 70 seconds, will the data analysis results only be shown if the irradiation direction of the laser sheet coincides with the downwind direction?

Reply: Thank you for pointing this out; this should be clarified in the manuscript.

Indeed the PIV measurements were only possible when the wind direction was aligned with the laser sheet. And yes, the reviewer is correct, the data analysis results are only shown when the direction of the laser sheet coincides with the downwind direction. The 70 second duration was not dictated by the wind direction. This imaging duration was set by the size of the computer memory (RAM), which was filled after 70 seconds of PIV acquisition. The memory was subsequently emptied as the images were transferred to hard drives; this operation lasted approximately 10 min. Only then was the next 70-second run possible. This resulted in a series of 70 s datasets (each acquired at 15 Hz), every 10 min. Additionally, we note that, in general, when the wind direction was not aligned with the laser sheet, there were no seeding particles within the PIV images, rendering any PIV measurements of the airflow impossible.

We have added the following text to the manuscript:

"Prior to starting the PIV measurements, the dominant wind direction was monitored live using a sonic anemometer. Once the wind direction was approximately aligned with the laser sheet (direction

perpendicular to the starboard boom), the PIV data acquisition started. This procedure ensured the presence of seeding particles within the laser light sheet, and that the PIV technique captured the dominant wind velocity components."

Comment 2 In order to calculate the growth rate of the wind wave, it is necessary to define the friction velocity. Firstly, is a constant flux layer confirmed?

Reply: Thank you for this comment. We agree that the friction velocity estimate method should be specified.

The friction velocity used here to compute the growth rate ($u_* \sim 0.3$ m/s), was estimated by applying the COARE bulk flux algorithm (Edson et al. [2013], Fairall et al. [1996, 2003]) to the wind velocity measured at a 10.6 m height (weather station). This measurement was chosen because it was our highest (above the water level) wind velocity measurement on the starboard boom, thus likely minimizing the effect of the waves on the estimate. Ortiz-Suslow et al. [2021] performed wind speed measurements (vertical profiles of the wind speed) on the port boom of FLIP, during the same measurement campaign. Their study suggests the existence of a constant flux layer at the time of our measurements, with an estimated friction velocity of ($u_* \sim 0.29$ m/s) [Ortiz-Suslow et al., 2021].

We have added the following text to the methods section of the manuscript:

"The latter measurement was used to estimate the friction velocity (u_), needed for the calculations of wave age and growth rate, via the COARE bulk flux algorithm (Edson et al. [2013], Fairall et al. [1996, 2003]). During the same time period, Ortiz-Suslow et al. [2021] also performed wind speed measurements (vertical profiles of the wind speed), on the port boom of FLIP. Their observations point to the existence of a constant flux layer at the time of our measurements, and are in good agreement with our total momentum flux estimates [Ortiz-Suslow et al., 2021]."*

Comment 3 If a data sample is in the 70s, is statistical representativeness guaranteed? And how has it been confirmed?

Reply:

Thank you for this comment. We recognize that this relatively short acquisition duration can be concerning. We have added the following to the methods section:

"Each 70 s dataset represents a total of 1050 2D velocity fields, each with up to 6776×2301 measured velocity vectors, or a total of 2,416,050 (vertical) velocity profiles."

For the slow wave results, each phase average value (figure 4, panels a and b) results from an average over at least 2507 instantaneous velocity measurements. The total number of slow waves is estimated to be over 2000. The maximum 95% confidence interval shows no significant difference from the average over all data (see figure R 1).

A similar analysis was performed for the previous 70 s PIV dataset (acquired approximately 10 min prior to the one presented in this study, due to hardware restrictions, as mentioned above). The wind and wave conditions were similar ($U_{10} = 7.9$ m/s, $u_* = 0.29$ m/s, $C_p = 10$ m/s) as in the present study ($U_{10} = 8.3$ m/s, $u_* = 0.3$ m/s, $C_p = 10.4$ m/s). The results show no significant difference with respect to the phase averaged contours of the horizontal and vertical velocity components (see figure R 2).

For the fast waves, the total number of waves sampled within 70 s is only approximately 11 (see figure S3, in the supplementary material). However, the standard deviation is low, as these long, fast waves consistently force the airflow in a quasi linear way. This is confirmed by the confidence interval figure

Figure R 1: Mean wave-coherent airflow velocities, confidence interval

below (figure R 3), where no significant difference is visible. Again, a comparison with the conditions 10 min prior, shows that the shown results are robust (figure R 4).

Comment 4 Comparison with previous research

In order to make the validity of the PIV measurements more robust, it is recommended to compare the results of this study with the results of previous research. For example, Sullivan et al. (2000, JFM) show the turbulent intensity and Reynolds stress distribution at the wavy boundary obtained by DNS. It would be better to show to what extent the turbulent characteristics obtained in this study agree with them.

Reply: Thank you for this suggestion. Regarding the turbulence, the computation of the turbulent stress tensor components from the PIV within the wave boundary layer requires several additional data processing steps that are beyond the scope of this study. So far, we were not able to obtain satisfying results for this, likely because of the limited duration of the dataset.

However, we were able to compare the wave-coherent velocity fields with those of Sullivan et al. [2000]. For the fast wave results, a comparison to previous research is mentioned in the manuscript:

"This result is in general agreement with past single-point observations [Hristov et al., 2003, Grare et al., 2013a, Zippel et al., 2024], and numerical simulations [Sullivan et al., 2000, 2008].", where we have now also added the originally omitted reference to Sullivan et al. [2000].

For the slow waves, we have now added the following text to the manuscript:

"These in situ observations of wave-induced airflow velocities within the first millimeters above slow wind waves (with estimated wave ages c_p/u_ between 2.5 and 4.2) are in good agreement with*

Figure R 2: Mean wave-coherent airflow velocities, comparison with 10 min prior

past laboratory [Buckley and Veron, 2016, 2019] and numerical [Sullivan et al., 2000] studies. There, similar phase shifts between wave-induced perturbations and water surface elevation were observed and modeled, albeit in simplified wave conditions (short fetch and monochromatic, respectively). Notably, the presently observed phase shifts match closest with the numerical reference case $c_p/u_* = 0$ of Sullivan et al. [2000], for flow over a solid wavy surface. This further supports the hypothesis that the relatively fast airflow is decoupled from the slow waves' orbital motions, leading to sheltering as the dominant coupling mechanism."

Comment 5 As a conclusion of this study, it is difficult to understand the position of the Miles mechanism (1958) and the sheltering mechanism proposed by Jeffreys (1925) in the development of wind waves. While it is stated that the contribution of the sheltering mechanism to the wind wave growth rate is large, the importance of the Miles mechanism is claimed in the critical layer of a few millimetres.

Reply: Thank you for this comment. We recognize that this part is confusing and meddles with the clarity of the message. We have removed the following paragraph from the Results and Discussion section:

"It should nonetheless be noted that a critical layer in the vicinity of the top of the viscous sublayer can be dynamically significant, even for strongly wind-forced waves of similar age (see for example Carpenter et al. [2022], who found a significant role of the critical layer just 2 mm above the surface). Hence it is possible that the critical layer for the slow waves and its associated dynamical role are not fully resolved here, since the mean viscous sublayer thickness here is approximately 0.5-0.7 mm (law of the wall)."

We have re-written the Conclusion section and added the following text:

"For intermediate wave ages, (e.g., for $14 \lesssim c/u_* \lesssim 25$ [Belcher and Hunt, 1998]), the competing dynamical roles of turbulent (e.g., sheltering) versus linear (e.g., Miles' critical layer theory, [Miles, 1957]) mechanisms, remain a topic of active research. New methods are being developed, to quantify these processes from our high resolution airflow observations, within this previously inaccessible region of the atmosphere-ocean boundary layer."

Figure R 3: Mean wave-coherent airflow velocities, confidence interval

Reviewers 2 and 3

This paper presents field measurements of the airflow separation over short and long waves during a campaign with FLIP outside the coast of California. The results point to two regimes for how surface waves are coupled to the airflow. The short waves are observed to experience a sheltering mechanism, whereas the long (70m and more) affect the airflow. The results are important and the reporting is timely since the general problem of wave growth is one which has direct implications for numerical weather prediction and climate modeling. The problem remains far from solved, but there is general agreement that a critical layer theory is the most efficient mechanism by which momentum is transferred between the atmosphere and the wave field. The authors are thus presenting a somewhat contrarian view, in that the sheltering seems to be quite important. They also claim to observe a direct momentum transfer from the long waves to the lower atmosphere. This latter mechanism has been studied experimentally (see Semedo et al, 2009), and is potentially of importance for swell attenuation. The middle ground, however, is not well covered by this study. That is, we are presented with very long waves that outrun the local wind and very short waves that are strongly forced by the wind. The critical layer theory is supposedly filling in the intermediate scales, but the authors state that the critical layer falls either too close to the wavy surface to be resolved for their short waves or is too high up in the case of long waves. This is a weakness of the study since the community would benefit from knowing how well the critical layer theory

Figure R 4: Mean wave-coherent airflow velocities, comparison with 10 min prior

holds up in such experiments. We acknowledge that the extent to which this is feasible may be limited by the experimental setup, but we would encourage the authors to look for evidence of this middle ground where critical layer theory should apply. Moreover, it is not entirely clear from the manuscript how long the experiments went on; this should be clarified in the revisions.

Having said that, the paper is well written, relevant and important, and we would recommend publication following a round of revisions.

Major comments and questions

The literature review should be expanded to set the current experiments and results in context. This could be done either in the introduction or in the supplementary materials.

Reply: Thank you for this comment, we realize that the introduction was too specialized. We have now significantly expanded the literature review in the introduction. We have introduced both airflow separation dynamics over short, slow waves, and wave-induced airflow motions over long, fast waves, both relevant concepts for this study. We have also added details on the debate over wave growth mechanisms. We have mentioned very fast-running waves (swells) and their possible net upward momentum flux effect.

Can the analysis be extended to also cover mid-range wavelengths for which the critical layer would presumably coincide with the range of the PIV measurements?

Reply: Thank you for this question. This is ongoing work that has, so far, not been successful, in spite of many efforts in this direction. Other datasets (acquired a few hours before/after the presented dataset) contain conditions, where we expect the critical layer of the dominant waves to fall within the field of view of the PIV. However, we were not able to extract clear (converging), meaningful results for these conditions. We speculate that this is due to the presence of both sheltering and critical layer mechanisms, but the resolution of this issue will require new techniques that have not yet been developed. This is for future work and is now mentioned in the conclusions:

"For intermediate wave ages, (e.g., for $14 \lesssim c/u_ \lesssim 25$ [Belcher and Hunt, 1998]), the competing dynamical roles of turbulent (e.g., sheltering) versus linear (e.g., Miles' critical layer theory, [Miles, 1957]) mechanisms, remain a topic of active research. New methods are being developed, to quantify these processes from our high resolution airflow observations, within this previously inaccessible region of the atmosphere-ocean boundary layer."*

Using our 3D anemometer and wave gauge data, we were able to identify critical layers using cross-spectral analysis when the anemometer is at the critical height of the dominant waves, similarly to Hristov et al. [2003], Grare et al. [2013b], Wu et al. [2018], but during those times, the PIV field of view does not extend high enough to encompass the critical layer.

Since this study shows new, statistically robust results for sheltering past slow short waves, as well as new, 2D PIV velocity field-based results for the longer waves, we believe it is worth reporting here, even without providing an answer so far for other wind-wave regimes, such as intermediate wave ages.

Specific comments by section

Introduction (As above) More in depth information about existing theories and contradictions between competing theories would be helpful to set the context for the current analysis and its importance, especially since the journal (Nature Communications) has a wide interdisciplinary scope. Therefore many readers are unlikely to have a strong background in the subject. If word limits prohibit the introduction from being expanded significantly, a more in-depth review of the literature covering the critical layer and sheltering mechanisms could be added to the supplementary materials for interested readers.

Reply: Thank you for this comment. We agree and have added additional details on wave growth mechanisms in the introduction, and also more clearly referred to the recent reviews on the topic:

"Wind wave growth results from a difference in airflow pressure between the leeward and windward sides of the wave [Phillips, 1977, Hristov et al., 2003]. However, the mechanisms that trigger this pressure perturbation, remain the subject of debate [Pizzo et al., 2021, Ayet and Chapron, 2022, Sullivan and McWilliams, 2010]. Physical interpretations are largely based on one of two competing theories, "critical-layer theory" [Miles, 1957], a linear shear-flow instability mechanism that takes place within the critical layer (thin region in the airflow, where the mean airflow speed matches that of the waves), and the "sheltering" mechanism, a turbulent process analogous to turbulent boundary modulations over low hills [Belcher and Hunt, 1998], whereby slow, wind-forced waves create a sheltered zone (from the wind) on their lee-side [Buckley et al., 2020]¹.

Field observations [Hristov et al., 2003, Grare et al., 2013a, Zippel et al., 2024] and high resolution laboratory measurements [Carpenter et al., 2022] suggest that waves travelling faster than the wind very near the surface, but slower than the wind at ~ 10 -m height, receive a significant amount of energy through the critical layer mechanism. However, recent laboratory measurements show frequent, highly turbulent airflow separation or "separated sheltering" [Belcher and Hunt, 1993] events over strongly forced wind waves [Buckley et al., 2020]; these are incompatible with linear critical-layer theory. On the other hand, longer waves that travel faster than the wind, are expected to cause sheltering effects on their windward side [Belcher and Hunt, 1998] (though observational evidence is scarce), and very fast-running waves (swells) can return momentum back to the atmosphere in the form of wave-induced

¹For a more extensive review of wind-wave coupling mechanisms, the reader is referred to reviews by Pizzo et al. [2021], Sullivan and McWilliams [2010], and Ayet and Chapron [2022].

winds [Semedo et al., 2009, Hanley and Belcher, 2008, Sullivan et al., 2008, Smedman et al., 1999, Grachev and Fairall, 2001].

When the wind forcing increases, wave steepening, breaking, and related turbulent processes are associated with an increase in air-sea exchanges of momentum and scalars [Sullivan and McWilliams, 2010, Edson et al., 2013, Veron, 2015, Richter and Veron, 2016, Brumer et al., 2017, Deike and Melville, 2018]. In high to very high wind conditions, momentum and CO₂ fluxes across the ocean surface eventually stop increasing [Powell et al., 2003, Donelan et al., 2004, French et al., 2007, Curcic and Haus, 2020, Bell et al., 2013]. Airflow separation past wave crests has been suggested as a possible reason for this flux saturation [Donelan et al., 2004, Janssen and Bidlot, 2023, Bell et al., 2013]. Yet direct, in situ observational evidence of airflow separation is lacking, due to difficulties in measuring airflow within centimeters of the wavy ocean surface."

The dataset was collected already in 2017. Is this the first paper to analyze the dataset, or have there been other publications using the same data?

Reply: Thank you for pointing this out. We agree that this is a long time for a first publication on these exciting measurements. This is the first manuscript to analyze the dataset. There are no other publications using this data. The relatively long delay is linked to the challenges that came with the data processing methods developed for this study. These novel, in situ airflow PIV measurements over the ocean waves, required the development of new image and data processing methods tailored to these field measurements. And, as mentioned above, we did spend a considerable amount of time and effort looking for critical layers and associated airflow velocity phase shifts within the PIV measurements, but, so far, without success.

2.1 Airflow separation events Fig. 2: Does the green laser penetrate the water surface? Do the gray regions beneath the air-sea interface in Fig. 2 represent the water column under the surface, or is it a re-projection of the surface reflection?

Reply: Thank you for this question; we recognize that these types of images can be confusing, and that the manuscript benefits from explaining this better. The green laser light penetrates the water surface, creating a light source below the surface. This light is then refracted through the rough water surface. Therefore, on the images, the gray regions that are visible beneath the air-sea interface, do not represent the water column under the surface, they are a representation of the slope of the portion of the sea surface that is between the camera and the laser sheet. The representation remains qualitative because this portion of the image is not within the focal plane of the camera lens. We realize that this was unclear in the text and have added the following text to the methods section:

"As the laser light penetrates the water surface, the intersection of the laser sheet with the water surface delivers a high resolution (300 μm/pixel) picture of the instantaneous geometry of the water surface (upper boundary of the gray scale region in Fig. 2). The intensity variations below the interface are caused by the refraction of green PIV laser light through the rough water surface, and are as such a function of the local variations in the slope of the water surface [see also Liu and Duncan, 2003, Duncan et al., 1999]."

Fig. 2: Is the example short wave riding the crest or trough of a long wave? One would assume the former based on the text in the Methods section, but it would be helpful to state this more clearly also in the main text. Are the axes rotated relative to the instantaneous long-wave slope?

Reply: Thank you, this was indeed not very clear in the text. Yes, the short wave is at the crest of a longer wave. We have added the following to the caption of figure 2:

"The age of this wave (with a wavelength ~ 1 m) is $c/u_ = 6.4$ or $c/U_{10} = 0.22$, which classifies it as a "slow" wave with respect to the wind speed blowing over it [Belcher and Hunt, 1998]. It is riding on the crest of a longer wave (with a wavelength ~ 70 m)."*

The images are all projected into earth coordinates (vertical z-axis). We would not expect any significant changes in the results, were the axes rotated relative to the instantaneous long-wave slope, because the long-wave slopes remain small.

2.2 Instantaneous structure of the airflow over fast waves Fig. 3: Are both the wave crest and (mean) wind propagating from left to right? Is windward/leeward defined with respect to the crest velocity? This could be specified in the caption.

Reply: Thank you, this was indeed confusing. We have added the following to the figure caption:

"Both the mean wind, and the wave, with a wavelength ~ 70 m, are propagating from left to right."

And we have added some clarification on windward/leeward in the text:

"Contrary to the slow wave case where the sheltered region is to the right of the crest (leeward side with respect to the mean wind), the magnitude of the horizontal velocity (panel a) decreases here on the left side of the wave, which is travelling faster than the local, near-surface wind."

2.3 Wave-induced motions How many individual slow and fast waves were used to generate the ensemble-averaged airflow profiles? What is the full duration of the experimental dataset?

Reply: Thank you for asking this. As mentioned to Reviewer 1, we recognize that this relatively short acquisition duration can be concerning. The duration of the presented dataset is 70 seconds. This imaging duration was set by the size of the computer memory (RAM), which was filled after 70 seconds of PIV acquisition. The memory was subsequently emptied as the images were transferred to hard drives; this operation lasted approximately 10 min. Only then was the next 70-second run possible. This resulted in a series of 70 s datasets (each acquired at 15 Hz), every 10 min.

We have added the following to the methods section:

"Each 70 s dataset represents a total of 1050 2D velocity fields, each with up to 6776×2301 measured velocity vectors, or a total of 2,416,050 (vertical) velocity profiles."

For convenience, we have also copied part of our response to Reviewer 1 here:

For the slow wave results, each phase average value (figure 4, panels a and b) results from an average over at least 2507 instantaneous velocity measurements. The total number of slow waves is estimated to be over 2000. The maximum 95% confidence interval shows no significant difference from the average over all data (see figure R 1).

A similar analysis was performed for the previous 70 s PIV dataset (acquired approximately 10 min prior to the one presented in this study, due to hardware restrictions, as mentioned above). The wind and wave conditions were similar ($U_{10} = 7.9$ m/s, $u_* = 0.29$ m/s, $C_p = 10$ m/s) as in the present study ($U_{10} = 8.3$ m/s, $u_* = 0.3$ m/s, $C_p = 10.4$ m/s). The results show no significant difference with respect to the phase averaged contours of the horizontal and vertical velocity components (see figure R 2).

For the fast waves, the total number of waves sampled within 70 s is only approximately 11 (see figure S3, in the supplementary material). However, the standard deviation is low, as these long, fast waves consistently force the airflow in a quasi linear way. This is confirmed by the confidence interval figure

below (figure R 3), where no significant difference is visible. Again, a comparison with the conditions 10 min prior, shows that the shown results are robust (figure R 4).

Do the slow wave ensemble averages contain breaking waves? If not, how do (or would) the mean airflow profiles differ for breaking waves?

Reply: In the presented dataset, we do not observe any air-entraining breaking waves. At other times outside of this dataset, we have observed intense separated sheltering events (airflow separation) past the crest of slow, air-entraining breaking waves. (Unfortunately the quality of the PIV measurement is reduced above such waves, due to intense light reflections.) Based on our past work in wind-wave tunnels, where we have measured airflow dynamics over breaking and non-breaking waves, we would expect similar results for the wave-induced perturbations over breaking waves, see for example Buckley and Veron [2019], figure 4, panels f and i. There, with a mean wind speed of 16.63 m/s, we expect frequent wave breaking events. Interestingly, in spite of dramatic airflow separation events (with flow reversal in the sheltered region), the average flow is generally not reversed. This result contrasts with results from numerical simulations over monochromatic waves or artificially narrow-banded breaking wave conditions [Yang and Shen, 2010, Sullivan et al., 2018], where phase averages show flow reversal in the average. We should mention here that airflow separation events over real wind waves (including laboratory waves) are very turbulent, intermittent events, and with a broad range of spatial scales (the sheltered area varies rapidly in space and time). We have found that phase averages of the airflow velocities do not generally represent this variability (in the laboratory as well as in situ).

Eq. 1: How were the inner layer heights L_i determined or defined?

Reply: Thank you for this question, we realize that this should have been detailed in the manuscript. The inner layer height was determined using equation 26 from Belcher and Hunt [1998], which equates the eddy advection time scale to that of eddy dissipation, leading to: $kL_i(U(L_i) - c) = 2\kappa u_*$, where κ is the von Karman constant, c the peak phase speed of the waves, k the wavenumber, $U(z)$ the mean wind velocity profile. We have added more details on this in the manuscript:

"which satisfies $kL_i(U(L_i) - c) = 2\kappa u_$, where κ is the von Karman constant, c the peak phase speed of the waves, k the wavenumber, and $U(z)$ the mean wind velocity profile (Belcher and Hunt [1998] (their equation 26, see also Grare et al. [2013a], Buckley and Veron [2016])"*

Fig. 4: (How) Does the long-wave modulation of the short waves affect the mean airflow patterns? Does the mean airflow of the slow waves cover all long-wave phases? Would the authors expect there to be a difference in the short-wave airflow at different long-wave phases due to long-wave modulation of the short waves (e.g. crest vs trough of long waves)?

Reply: Thank you, we have been looking into this. Yes we would expect a difference in short-wave airflow modulations at the crests of the long waves, compared with the troughs. Unfortunately the data quality is not sufficient, in locations of the airflow very near the surface above long wave troughs, to conclude anything on this at this time. In figure R 5, we show short wave phase averaged quantities, for 4 quadrants of long wave phases. We note here that the (near-surface) slow wave phase averages above the trough of the fast waves do not converge toward any meaningful results. We have raised the issue of insufficient seeding in the methods section of the manuscript.

Fig. 4: Alternate layout suggestion: Keep 2-column layout, but with the left column consisting of 4 subpanels in each row, and the right panel consisting of one high panel (current f panel). Left column: 4-panel column with c+d on top 2 rows of left column. Include the wave form in the same way as in a+b (so that all subpanels include 0 on the y axis). On 3rd and 4th row of left column,

Figure R 5: **Phase-averaged slow wave wave-coherent velocities**, at 5 different phase ϕ ranges of the fast waves: From left to right, $-\pi < \phi < -\pi/2$, $-\pi/2 < \phi < 0$, $0 < \phi < \pi/2$, $\pi/2 < \phi < \pi/2$, $-\pi < \phi < \pi$. N is the number of non-nan data values going into each average value (1 pixel of one colormap).

show current a+b panels, with e.g. a box showing the horizontal and vertical scales of the short waves vs the long waves.

Reply: Thank you for this suggestion. We agree that the layout of figure 4 could be improved. We have now separated this figure into 2 figures, and modified the layout of the wave-coherent velocity plots. This now shows better the significant differences between slow wave and fast wave-coherent velocities. Our updated figure(s) do not quite follow the reviewer's suggestion for 2 reasons: First, we are not sure how to design a box with to-scale comparison of waves with (H, λ) scales of $O(1, 100)m$ vs $O(1, 100)cm$. Second, we should have mentioned that the very near-surface wave-coherent velocities over the slow waves shown in figure 4, are computed in a surface-following frame of reference, before being transformed back into cartesian coordinates for this plot. For the more regular fast waves, the averaging was performed in the cartesian frame of reference. There, due to variability in wave heights and in PIV seeding density near the surface, and especially just above wave troughs, we have low confidence in the averaged near surface values. We have included these here for the reviewer's convenience (figure R 6).

We have added the following text to the methods section of the manuscript:

"In the case of the slow waves [...], The velocity fields were mapped from the cartesian frame of reference (x, z) to a surface-following frame of reference (x, ζ) , where $\zeta = z - \eta(x, t)$ (as was previously done in laboratory studies, for example by Grare et al. [2013b], Tenhaus et al. [2024]). For the comparatively more regular, fast waves, the phase averaging was performed in the cartesian frame of reference. There, due to variability in wave heights and in PIV seeding density near the surface, and especially just above wave troughs, we have low confidence in the averages below $z = 0.6 m$. Therefore these near surface averages are not shown here (figure 4c-d)."

Reviewing this figure has also allowed us to identify an error (offset) in the vertical axis of the fast wave plots (figure 4, c-d), as well as in the sheltering wave growth estimate (now figure 5), which we have now fixed.

Figure R 6: Mean wave-coherent airflow velocities

3 Conclusion The authors summarize their results, which show that short/slow waves are coupled to a sheltering/separation response in the nearfield airflow, whereas the long/fast waves induce orbital wakes in the airflow. This result is essentially (redundantly) stated twice back-to-back. What is lacking for readers with basic (or limited) background in the subject is a clear statement of how the presented results change or update our understanding of the process of wave growth by wind. As stated above, this should preferably be explained more thoroughly in the introduction, but more details could be added to the Supplementary materials.

Reply: Thank you for this comment. We have removed one instance of the redundant statement and we have modified both introduction and conclusion, following the reviewers' previous comments.

4.1 In situ Large field of view Particle Image Velocimetry (PIV) Will the code for the PIV algorithm used in this study be made publicly available?

Reply: Thank you for this question. It is not planned to publish the PIV algorithm for this study. We use an in-house code, but it is not particularly interesting. In fact we do not expect the results to be PIV-algorithm dependent. We expect that commercially available PIV software (for example from Dantec or Lavis) would produce similar instantaneous 2D velocity fields, provided that some important pre and post-processing steps are taken (e.g., image stitching, proper masking of areas without seeding, motion correction).

What level smoothing or filtering was performed on the raw PIV velocities vs. the final results shown in the manuscript?

Reply: Figures 2 and 3 in the manuscript show direct outputs of the PIV velocity computation algorithm. Within the final step of the PIV algorithm, velocity vectors with low interrogation window cross-correlation values (less than 50%) are removed and replaced using a 2D robust spline smoothing algorithm.

4.2 Environmental conditions and supporting measurements Are directional wave spectra available for the measurement campaign? These would be helpful for showing the relative directionality of the windsea and swell.

Reply: Thank you, we have considered this. A Datawell directional waverider buoy (CDIP #234, https://cdip.ucsd.edu/themes/cdip/news?pb=1&d2=p9&u3=v:sensors:p_id:p9:mode:all:s:234:st:1) was moored within 1 km of FLIP's location. Directional wave spectra show that the dominant waves were aligned with the wind on Oct 20th. On Oct 17th, there was additionally a 12 s swell system with a direction perpendicular to the wind sea. We have now specified the 1 km distance between the wave buoy and FLIP in the manuscript:

"Alignment of the peak wave travel direction with the local wind direction was checked using directional wave field information provided by a waverider buoy moored within 1 km of FLIP's location."

Will any of the data and/or code used in the manuscript be made publicly available?

Reply: Thank you. We plan to share the data that is shown in all the figures, including the supplementary ones.

References

References

- J B. Edson, V Jampana, R A Weller, S P Bigorre, A J Plueddemann, C W Fairall, S D Miller, L Mahrt, D Vickers, and H Hersbach. On the Exchange of Momentum over the Open Ocean. *Journal of Physical Oceanography*, 43(8):1589–1610, August 2013. ISSN 0022-3670. doi: 10.1175/JPO-D-12-0173.1. URL <http://journals.ametsoc.org/doi/abs/10.1175/JPO-D-12-0173.1>.
- C W Fairall, E F Bradley, D P Rogers, J B Edson, and G S Young. Bulk parameterization of air-sea fluxes for tropical ocean-global atmosphere coupled-ocean atmosphere response experiment. *J. Geophys. Res.*, 101:3747–3764, 1996.
- C W Fairall, E F Bradley, J E Hare, A A Grachev, and J B Edson. Bulk parameterization of air-sea fluxes: Updates and verification for the coare algorithm. *J. Climate*, 16(4):571–591, 2003.
- David G Ortiz-Suslow, John Kalogiros, Ryan Yamaguchi, and Qing Wang. An evaluation of the constant flux layer in the atmospheric flow above the wavy air-sea interface. *Journal of Geophysical Research: Atmospheres*, 126(8):e2020JD032834, 2021.
- P P Sullivan, J C McWilliams, and CH Moeng. Simulation of turbulent flow over idealized water waves. *J. Fluid Mech.*, 404:47–85, 2000. doi: 10.1017/S0022112099006965.
- TS Hristov, SD Miller, and CA Friehe. Dynamical coupling of wind and ocean waves through wave-induced air flow. *Nature*, 422(6927):55–58, 2003.
- L. Grare, L. Lenain, and W. K. Melville. Wave-Coherent Airflow and Critical Layers over Ocean Waves. *Journal of Physical Oceanography*, 43(10):2156–2172, 2013a. doi: 10.1175/JPO-D-13-056.1. URL <http://journals.ametsoc.org/doi/abs/10.1175/JPO-D-13-056.1>.

- Seth F Zippel, James B Edson, Malcolm E Scully, and Oaklin R Keefe. Direct observation of wave-coherent pressure work in the atmospheric boundary layer. *Journal of Physical Oceanography*, 54(2):445–459, 2024.
- Peter P. Sullivan, James B. Edson, Tihomir Hristov, and James C. McWilliams. Large-eddy simulations and observations of atmospheric marine boundary layers above nonequilibrium surface waves. *Journal of the Atmospheric Sciences*, 65:1225–1245, 2008. doi: 10.1175/2007JAS2427.1.
- Marc P Buckley and Fabrice Veron. Structure of the airflow above surface waves. *Journal of Physical Oceanography*, 46(5):1377–1397, 2016.
- M P Buckley and F Veron. The turbulent airflow over wind generated surface waves. *European Journal of Mechanics*, 58(11):161, 2019.
- JR Carpenter, MP Buckley, and F Veron. Evidence of the critical layer mechanism in growing wind waves. *Journal of Fluid Mechanics*, 948:A26, 2022.
- S E Belcher and J C R Hunt. Turbulent flow over hills and waves. *Annu. Rev. Fluid Mech.*, 30:507–538, 1998. ISSN 0066-4189.
- J Miles. On the generation of surface waves by shear flows. *J. Fluid Mech.*, 3:185–204, 1957.
- L Grare, W L Peirson, Hu Branger, J W Walker, J-P Giovanangeli, and V Makin. Growth and dissipation of wind-forced, deep-water waves. *J. Fluid Mech.*, 722:5–50, 2013b. doi: 10.1017/jfm.2013.88. URL http://journals.cambridge.org/abstract/_S0022112013000888.
- Lichuan Wu, Tihomir Hristov, and Anna Rutgersson. Vertical profiles of wave-coherent momentum flux and velocity variances in the marine atmospheric boundary layer. *Journal of Physical Oceanography*, 48(3):625–641, 2018. doi: 10.1175/JPO-D-17-0052.1. URL <https://doi.org/10.1175/JPO-D-17-0052.1>.
- O M Phillips. *The Dynamics of the Upper Ocean*. Cambridge University Press, 1977.
- Nick Pizzo, Luc Deike, and Alex Ayet. How does the wind generate waves? *Physics Today*, 74(11):38–43, 2021.
- Alex Ayet and Bertrand Chapron. The dynamical coupling of wind-waves and atmospheric turbulence: a review of theoretical and phenomenological models. *Boundary-Layer Meteorology*, 183(1):1–33, 2022.
- P P Sullivan and J C McWilliams. Dynamics of winds and currents coupled to surface waves. *Annu. Rev. Fluid Mech.*, 42:19–42, 2010.
- MP Buckley, F Veron, and K Yousefi. Surface viscous stress over wind-driven waves with intermittent airflow separation. *Journal of Fluid Mechanics*, 905:A31, 2020.
- S E Belcher and J C R Hunt. Turbulent shear flow over slowly moving waves. *J. Fluid Mech.*, 251:119–148, 1993.
- Alvaro Semedo, Øyvind Saetra, Anna Rutgersson, Kimmo K Kahma, and Heidi Pettersson. Wave-induced wind in the marine boundary layer. *Journal of the Atmospheric Sciences*, 66(8):2256–2271, 2009.

- Kirsty E Hanley and Stephen E Belcher. Wave-driven wind jets in the marine atmospheric boundary layer. *Journal of the Atmospheric Sciences*, 65(8):2646–2660, 2008.
- A. Smedman, U. Högström, H. Bergström, A Rutgersson, K. K. Kahma, and H. Pettersson. A case study of air-sea interaction during swell conditions. *J. Geophys. Res.*, 104(C11):25833–25851, 1999.
- AA Grachev and CW Fairall. Upward momentum transfer in the marine boundary layer. *J. Phys. Oceanogr.*, 31(7):1698–1711, 2001.
- F Veron. Ocean spray. *Annu. Rev. Fluid Mech.*, 47:507–538, 2015. doi: 10.1146/annurev-fluid-010814-014651.
- David H Richter and Fabrice Veron. Ocean spray: An outsized influence on weather and climate. *Physics Today*, 69(11):34–39, 2016.
- Sophia E Brumer, Christopher J Zappa, Byron W Blomquist, Christopher W Fairall, Alejandro Cifuentes-Lorenzen, James B Edson, Ian M Brooks, and Barry J Huebert. Wave-related reynolds number parameterizations of co2 and dms transfer velocities. *Geophysical Research Letters*, 44(19):9865–9875, 2017.
- Luc Deike and W Kendall Melville. Gas transfer by breaking waves. *Geophysical Research Letters*, 45(19):10–482, 2018.
- M D Powell, P J Vickery, and T A Reinhold. Reduced drag coefficient for high wind speeds in tropical cyclones. *Nature*, 422:279–283, Mar 20 2003. doi: 10.1038/nature01481.
- M A Donelan, B K Haus, N Reul, W J Plant, M Stiassnie, H C Graber, O B Brown, and E S Saltzman. On the limiting aerodynamic roughness of the ocean in very strong winds. *Geophys. Res. Lett.*, 31(18), Sep 28 2004. doi: 10.1029/2004GL019460.
- J R French, W M Drennan, J A Zhang, and P G Black. Turbulent fluxes in the hurricane boundary layer. part i: momentum flux. *J. Atmos. Sci.*, 64:1089–1102, 2007. doi: 10.1175/JAS3887.1.
- Milan Curcic and Brian K Haus. Revised estimates of ocean surface drag in strong winds. *Geophysical research letters*, 47(10):e2020GL087647, 2020.
- TG Bell, W De Bruyn, SD Miller, B Ward, KH Christensen, and ES Saltzman. Air–sea dimethylsulfide (dms) gas transfer in the north atlantic: evidence for limited interfacial gas exchange at high wind speed. *Atmospheric Chemistry and Physics*, 13(21):11073–11087, 2013.
- Peter AEM Janssen and Jean-Raymond Bidlot. Wind–wave interaction for strong winds. *Journal of Physical Oceanography*, 53(3):779–804, 2023.
- X Liu and J H Duncan. The effects of surfactants on spilling breaking waves. *Nature*, 421:520–523, 2003.
- JH Duncan, H Qiao, V Philomin, and A Wenz. Gentle spilling breakers: crest profile evolution. *J. Fluid Mech.*, 379:191–222, 1999. URL http://journals.cambridge.org/abstract/_S0022112098003152.

- D Yang and L Shen. Direct-simulation-based study of turbulent flow over various waving boundaries. *J. Fluid Mech.*, 650:131–180, 2010.
- Peter P. Sullivan, Michael L. Banner, Russel P. Morison, and William L. Peirson. Turbulent flow over steep steady and unsteady waves under strong wind forcing. *Journal of Physical Oceanography*, 48(1):3–27, 2018. doi: 10.1175/JPO-D-17-0118.1. URL <https://doi.org/10.1175/JPO-D-17-0118.1>.
- Janina Tenhaus, Marc P Buckley, Silvia Matt, and Ivan B Savelyev. Viscous and turbulent stress measurements above and below laboratory wind waves. *Experiments in Fluids*, 65(12):174, 2024.